# PC-X: Profound Clustering via Slow Exemplars

Yuangang Pan[1,2], Yinghua Yao[1,2], Ivor Tsang[1,2,3]

[1]Center for Frontier AI Research, Agency for Science, Technology, and Research,
[2]Institute of High Performance Computing, Agency for Science, Technology, and Research,
[3]School of Computer Science and Engineering, Nanyang Technological University
{yuangang.pan,eva.yh.yao,ivor.tsang}@gmail.com

Deep clustering aims at learning clustering and data representation jointly to deliver clustering-friendly representation. In spite of their significant improvements in clustering accuracy, existing approaches are far from meeting the requirements from other perspectives, such as universality, interpretability and efficiency, which become increasingly important with the emerging demand for diverse applications. We introduce a new framework named Profound Clustering via slow eXemplars (PC-X), which fulfils the above four basic requirements simultaneously. In particular, PC-X encodes data within the auto-encoder (AE) network to reduce its dependence on data modality (*universality*). Further, inspired by exemplar-based clustering, we design a Centroid-Integration Unit (CI-Unit), which not only facilitate the suppression of sample-specific details for better representation learning (*accuracy*), but also prompt clustering centroids to become legible exemplars (*interpretability*). Further, these exemplars are calibrated stably with mini-batch data following our tailor-designed optimization scheme and converges in linear (*efficiency*). Empirical results on benchmark datasets demonstrate the superiority of PC-X in terms of universality, interpretability and efficiency, in addition to clustering accuracy. The code of this work is available at `https://github.com/Yuangang-Pan/PC-X/`.

## 1. Introduction

Clustering is one of the most important techniques for exploring data structures in an unsupervised manner. The objective of clustering is to partition samples into groups such that samples in the same group are more similar than that from different groups. Naively attempting to cluster samples using raw features [1–3] may not produce pleasing partitions as semantically similar samples are not necessarily close in the high dimensional feature space [4–6] the sample resides in.

Representation learning in deep learning has paved the way for obtaining a good data representation, usually of low dimensionality [7, 8]. Inspired by this, deep clustering suggests joint learning of representation and clustering within a single framework, typically via introducing a clustering loss over latent representations delivered by pre-trained models [9] or auto-encoders [10–12]. Benefiting from better data representations, these algorithms have reported significant performance gains on various benchmark datasets. Next, researchers in self-supervised learning (SSL) community take over the task and constantly refresh the state-of-the-art (SOTA) clustering records on various image datasets [5, 13–15].

Although deep learning based clustering approaches greatly facilitate the development of clustering analysis, such improvement is mainly limited to the aspect of clustering accuracy. Meanwhile, the development of clustering approaches in other aspects, e.g., universality, interpretability and efficiency, has remained stagnant or been degenerate over the past few years. Particularly, despite keeping the best records on many images datasets, clustering approaches based on SSL highly depend on the choice of invariant transformations and degenerate significantly if effective transformations are not available, like text data [5, 14]. Furthermore, jointly optimizing representation learning and clustering may incur efficiency concerns, since a large/full batch data is usually required to update the clustering structure for sake of stability [16–18]. Last but not least, most deep clustering algo-

First Conference on Parsimony and Learning (CPAL 2024).

rithms are deemed as black-box with less interpretability, as their criteria for clustering is unknown, which may not be consistent with samples' semantics, and even beyond human understanding [19].

In this paper, we design a new end-to-end framework called Profound Clustering via slow eXemplars (PC-X), which is inherent interpretable and universally applicable to various types of large-scale datasets. To be specific, instead of adding the clustering module on top of latent representation, PC-X incorporates it within the AE network via the proposed Centroid-Integration Unit (CI-Unit), which suppresses (clustering nonrelevant) sample-specific details and facilitates clustering centroids to become a set of exemplars through decoding. Regarding the inference, we decompose the optimization of representation learning and clustering into two sub-tasks by introducing an auxiliary variable. Given the representation learning, the exemplars, i.e., clustering centroids, can be calibrated slowly with exponential moving average; while given the exemplars, the representation of each sample gradually gathers around its respective clustering centroid through stochastic gradient optimization. In summary, the main contributions of this paper are summarized as follows:

- We introduce a new clustering framework (PC-X), which is inherent interpretable and universally applicable to various types of datasets. Extensive experiments on seven benchmark datasets demonstrates that PC-X outperforms existing clustering methods from four aspects, including clustering accuracy, universality, interpretability and efficiency.

- We introduce a new module call Centroid-Integration Unit, which not only helps to suppress (clustering nonrelevant) sample-specific details in the latent embedding, but also facilitates clustering centroids to become legible exemplars through decoding.

- We design an efficient optimization algorithm for PC-X, which updates all parameters in PC-X stably with mini-batch data, and enjoys superior convergence as well as algorithm complexity.

## 2. Problem statement and literature review

In this section, we first introduce the problem setting of clustering. Then, we summarize classical clustering as well as deep clustering methods, and discuss their deficiencies accordingly.

### 2.1. Cluster analysis

Let $X = \{x_n\}_{n=1}^N$ be a set of $N$ samples in $\mathbb{R}^D$, drawn from heterogeneous populations. Assume the cluster number of is known to be $K(\ll N)$. The goal of clustering is to partition $X$ into $K$ non-overlap groups $\mathcal{S} = (S_1, S_2, \ldots, S_K)$ with low intra-class variance but high inter-class variance.

The following four challenges should be reckoned when designing the clustering algorithm, namely

- **Accuracy** is the primary target. The designed algorithm is expected to partition samples according to the intrinsic semantic differences within the data. A good clustering algorithm should achieve consistent superior performances in terms of various clustering measures.

- **Universality** requires the clustering method is generally applicable to various types of data, e.g., image or text. While adapting the network architectures is necessary for different modalities, "universality" lies in the way that the method accommodates these architectures and can be effectively adapted to specific modalities based on a minimum knowledge of target data.

- **Interpretability** pertains to whether the criteria followed by clustering are easily interpretable for users. We argue that AE/DGM-based clustering methods inherently possess interpretability for their clustering results. Through the decoder, the clustering centroids can be mapped to the data space, allowing for the visualization of the clustering semantics (Fig. 2).

- **Efficiency** means that the algorithms have low memory space and low computational costs. Typically, updating deep clustering models stably requires large batch data, which is memory-inefficient for large-scale applications. Further, those methods also suffer from slow convergence due to a lack of proper joint optimization schemes.

Table 1: Comparison of various clustering paradigms in terms of clustering properties.

| Challenges | Classical clustering | Deep clustering | | | |
| --- | --- | --- | --- | --- | --- |
| | | Pre-trained model | AE/DGM | Self-supervised learning | PC-X |
| Accuracy | ✗ | ✗ | ✗ | ✓ | ✓ |
| Universality | ✗ | ✓ | ✓ | ✗ | ✓ |
| Interpretability | ✓ | ✗ | ✓ | ✗ | ✓ |
| Efficiency | ✓ | ✗ | ✗ | ✗ | ✓ |

## 2.2. Comparison of existing clustering paradigms

Existing clustering methods can be roughly categorized into classical clustering and deep clustering; while deep clustering can be further divided into three branches in terms of the ways how they combine clustering and representation learning. See Table 1 for a summary of comparisons.

**Classical clustering** generally refers to the algorithms that group samples in the raw feature space. This kind of approaches are easy to interpretable and efficient. However, they are applied to limited types of data with well-defined features. In terms of complex data (e.g., image), it fails to achieve good results since semantically similar samples are not necessarily close in the raw feature space.

**Deep clustering based on pre-trained model** generalizes classical clustering for complex data through executing clustering modules over latent representations delivered by pre-trained CNNs or AEs. In spite of its simplicity, it is highly dependent on the original latent embedding due to the lack of joint training of representation and clustering. As a consequence, differential clustering losses are derived in DEC [16] and Deepcluster [9] to fine-tune the pre-trained model.

**Deep clustering based on auto-encoder (AE) / deep generative model (DGM)** propose to jointly train clustering modules with extra tasks, such as self-reconstruction or distribution matching [18–22]. It is shown to exhibit robustness to poor initial latent embedding since they can keep the diversity in latent space via reconstructing/generating the input samples. Prominent works are JULE [17], IDEC [20], VaDE [23] and DEPICT [18]. Nevertheless, the reconstruction loss tends to overestimate sample-specific details (e.g. textures and background) which are unrelated to semantics [24], degrading clustering performance. Meanwhile, it is memory inefficient for large-scale applications, since multiple iterations over the whole dataset are required for joint updating the clustering structure and network parameters reliably [17, 18, 20].

**Deep clustering based on self-supervised learning (SSL),** e.g., TCC [25] and DivClust [26], achieve SOTA clustering results on various image datasets. As semantics pairs are constructed for each sample through a collection of predefined semantics-invariant transformations, the performance of SSL-based clustering methods highly depends on the choice of invariant transformations, which requires rich domain knowledge and varies from dataset to dataset [27–34]. If no effective invariant transformations are available, their performance would degenerate significantly. Meanwhile, lack of informative gradient, SSL-based approaches usually suffer from slow convergence [5].

Last but not least, despite significant performance gains based on representation learning, the clustering process of deep clustering is less interpretable as the criteria used for clustering are unknown, which may not be consistent with samples' semantics, and even out of human understanding.

## 3. Profound Clustering via Slow Exemplars

Taking into consideration the weakness of existing clustering approaches, we are committed to introducing a profound clustering framework that can achieve superior performance in terms of four challenges stated in Section 2 simultaneously.

### 3.1. Deep auto-encoder as the backbone

**Universality.** According to Table 1, we choose deep AE as the backbone to reduce the dependence on the data modality. Let $f_\theta$ and $g_\phi$ denote the encoder and decoder, respectively. We adopt soft

$k$-means to perform clustering in the latent space for the sake of efficiency. The objective of PC-X is summarized as follows:

$$\mathcal{L}_{PC\text{-}X}(\boldsymbol{\Delta},\boldsymbol{\mu}) = \mathcal{L}_{AE}(\boldsymbol{\Delta}) + \eta_1 \mathcal{L}_{cluster}(\boldsymbol{\mu}) = \underbrace{\frac{1}{N}\sum_{n=1}^{N} D(x_n\|g_\phi(z_n))}_{Reconstruction\ loss} + \eta_1 \underbrace{\frac{1}{N}\sum_{n=1}^{N}\sum_{k=1}^{K}\delta_{nk}\|z_n-\mu_k\|_2^2}_{Clustering\ loss}, \quad (1)$$

where $z_n = f_\theta(x_n)$ denotes the latent embedding. $D(.\|.)$ is the reconstruction loss. $\boldsymbol{\Delta} = \{\theta,\phi\}$ stands for network parameters. $\eta_1$ is the balance factor. The intermediate variable $\delta_{nk} \in \{0,1\}$ denotes the group assignment, assigning each latent code $z_n$ to its closest clustering centroid, i.e.,

$$\lambda_{nk} = \frac{\exp\left(-\tau\|z_n-\mu_k\|_2^2\right)}{\sum_{i=1}^{K}\exp\left(-\tau\|z_n-\mu_i\|_2^2\right)}, \qquad \delta_{nk} = \begin{cases} 1 & k = \operatorname{argmax}_j \lambda_{nj} \\ 0 & \text{otherwise} \end{cases}, \quad (2)$$

where $k = 1,2,\ldots,K$ and $n = 1,2,\ldots,N$. It was claimed that encouraging the diversity among clustering centroids contributes to a stable clustering process [35, 36]. We therefore add an extra minimum entropy (ME) regularization term in Eq. (1), i.e., $\mathcal{L}_{ME}(\boldsymbol{\Delta}) = -\frac{1}{N}\sum_{n=1}^{N}\sum_{k=1}^{K}\lambda_{nk}\log\lambda_{nk}$ to promote clustering stability. Note $\mathcal{L}_{ME}$ is optimized only for the network parameters ($\boldsymbol{\Delta}$) which will not cause extra optimization difficulty. A burn-in period (#epoch $\leq 250$) is introduced for the ME regularization since the estimation is less informative initially. See Fig. 4 for its effects on PC-X.

### 3.2. Interpreting clustering centroids as exemplars via reconstruction

In AE/DGM-based clustering, the latent representations $z_n$ inevitably contain sample-specific details due to self-reconstruction, which is detrimental to clustering [24]. We argue that this is lack of the insight of clustering centroids since the clustering centroids have never been fully utilized during the reconstruction. Specifically, we introduce the Centroid-Integration Unit (CI-Unit) (shown in Fig. 1) to incorporate the centroids into the training of AE/DGM.

$$\tilde{z}_n = \sum_{k=1}^{K}\delta_{nk}\mu_k = \mu_{\{\operatorname{argmax}_j \lambda_{nj}\}}, \quad \hat{z}_n = h_\psi(z_n,\tilde{z}_n). \quad (3)$$

The CI-Unit constructs a new latent representation $\hat{z}_n$ that incorporates not only the original latent embedding $z_n$ but also its closest clustering centroid $\tilde{z}_n$. For simplicity, we implement $h_\psi$ by simple concatenation along with one extra fully connection layer.

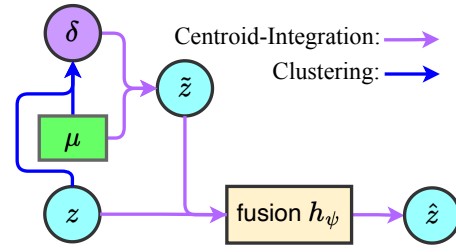

Figure 1: Centroid-Integration Unit

**Accuracy.** Our CI-Unit unifies both goals of clustering and reconstruction. This is because the variable $\mu$ should be the group-shared features if it can be incorporated to latent representation of each sample assigned to the same cluster for good reconstruction. As a result, the subsequently self-reconstruction using the centroid-enhanced representation $\hat{z}_n$ facilitates the suppression of sample-specific details in $z_n$, bringing a mutually reinforcing effect between accurate clustering centroid estimation and better self reconstruction. See A.6 for a comprehensive demonstration.

**Interpretability.** PC-X with CI-Unit possesses similar merits as exemplar-based clustering which minimizes a similar $k$-means objective, i.e., $\mathcal{L}(\boldsymbol{\mu}) = \frac{1}{N}\sum_{n=1}^{N}\sum_{k=1}^{K}\delta_{nk}D(x_n\|\mu_k)$, but restricts cluster centers (aka exemplars) to be chosen from training data (i.e., $\boldsymbol{\mu} \subseteq X$), and thus owns inherent interpretability [37]. Similarly, the reconstruction of our clustering centroids $g_\phi(h_\psi(\mu,\mu))$ would be enhanced to be legible *exemplars* since (1) each centroid-enhanced representation $h_\psi(f_\theta(x),\mu)$[1] is legitimately decoded to its respectively input $x$ during the whole training process (the reconstruction loss), i.e., $g_\phi(h_\psi(f_\theta(x),\mu)) \rightarrow x$; and (2) each latent representation is pushed close to its corresponding clustering centroid, i.e., $f_\theta(x) \rightarrow \mu$ (the clustering loss).

---

[1]$\mu$ refers to the closest clustering centroid of $x$, we omit the downscripts here for simplicity.

### 3.3. Stable and efficient clustering brought by slow exemplars

**Efficiency.** In Eq. (1), two types of parameters, i.e., network parameters $\boldsymbol{\Delta}$ and clustering parameters $\boldsymbol{\mu}$, are coupled together which hinders them from jointly optimizing. Different from network parameters $\boldsymbol{\Delta}$, the update of centroids $\boldsymbol{\mu}$ is unstable when optimized with stochastic gradient. This is because the centroids estimated on different mini-batches may differ markedly. In addition, when the centroids are estimated through standard full-batch iterations, it would lead to significant computational cost and storage overhead, especially for large-scale applications. Therefore, we propose to slowly calibrate the reserved centroids (i.e., exemplars) using mini-batch data instead of replacing them with the new estimations, which thus enjoys both stability and efficiency.

#### 3.3.1. Decomposing the model parameters for stochastic optimization

In this section, we propose a decomposition-coordination optimization method following the principle of alternating direction method of multipliers [38]. To be specific, we decompose the optimization regarding two types of parameters through introducing an auxiliary variable $\boldsymbol{c}$. Namely,

$$\mathcal{L}(\boldsymbol{\Delta}, \boldsymbol{c}, \boldsymbol{\mu}) = \mathcal{L}_{PC\text{-}X}(\boldsymbol{\Delta}, \boldsymbol{c}) + \rho \sum_{k=1}^{K} \|c_k - \mu_k\|_2^2, \tag{4}$$

where the penalty parameter $\rho$ denotes the level of model robustness against the mini-batch update.

Accordingly, our decomposition-coordination optimization method consists of the three iterations:

$$\boldsymbol{\Delta}^{t+1} = \underset{\boldsymbol{\Delta}}{\operatorname{argmin}} \, \mathcal{L}\left(\boldsymbol{\Delta}, \boldsymbol{c}^t, \boldsymbol{\mu}^t\right), \; \boldsymbol{c}^{t+1} = \underset{\boldsymbol{c}}{\operatorname{argmin}} \, \mathcal{L}\left(\boldsymbol{\Delta}^{t+1}, \boldsymbol{c}, \boldsymbol{\mu}^t\right), \; \boldsymbol{\mu}^{t+1} = \underset{\boldsymbol{\mu}}{\operatorname{argmin}} \, \mathcal{L}\left(\boldsymbol{\Delta}^{t+1}, \boldsymbol{c}^{t+1}, \boldsymbol{\mu}\right).$$

where $t$ is the index of epoch.

We then derive efficient update solutions for each subproblem, enabling stochastic update.

**In terms of $\boldsymbol{\Delta}$,** the reduced subproblem is a three objective optimization problem $\mathcal{L}_{PC\text{-}X}(\boldsymbol{\Delta}, \boldsymbol{c}^t)$ (set $\boldsymbol{\mu}$ to $\boldsymbol{c}^t$ in Eq. (1)). Since no closed-form solution exists, we optimize it with gradient descent,

$$\boldsymbol{\Delta}^{t+1} = \boldsymbol{\Delta}^t - \gamma \frac{\partial \mathcal{L}\left(\boldsymbol{\Delta}, \boldsymbol{c}^t, \boldsymbol{\mu}^t\right)}{\partial \boldsymbol{\Delta}}\bigg|_{\boldsymbol{\Delta}=\boldsymbol{\Delta}^t}, \tag{5}$$

where $\gamma$ is the step size. Eq. (5) can be implemented with advanced gradient descent approaches.

**In terms of $\boldsymbol{c}$,** the subproblem is a two-objective optimization problem. By setting the gradient of the objective to zeros, we can get an analytic solution.

$$\frac{\partial \mathcal{L}\left(\boldsymbol{\Delta}^{t+1}, \boldsymbol{c}, \boldsymbol{\mu}^t\right)}{\partial c_k} = 0 \Longrightarrow c_k = \frac{\rho \mu_k^t + \frac{\eta_1}{N} \sum_{n=1}^{N} \delta_{nk} z_n}{\rho + \frac{\eta_1}{N} \sum_{n=1}^{N} \delta_{nk}}, \quad k = 1, 2, \ldots, K. \tag{6}$$

Let $\kappa = \frac{\rho}{\rho + \frac{\eta_1}{N} \sum_{n=1}^{N} \delta_{nk}}$ and $\bar{\mu}_k^t = \frac{\sum_{n=1}^{N} \delta_{nk} z_n}{\sum_{n=1}^{N} \delta_{nk}}$, we have $c_k^{t+1} = \kappa \mu_k^t + (1 - \kappa)\bar{\mu}_k^t$, $k = 1, 2, \ldots, K$, where the auxiliary variable $\boldsymbol{c}$ pursues a balance between the exiting centroid $\boldsymbol{\mu}^t$ and the re-estimated one $\bar{\boldsymbol{\mu}}^t$. Eq. (6) actually is equivalent to the well-known exponential moving average (EMA).

**In terms of $\boldsymbol{\mu}$,** the reduced subproblem consists of the second item in Eq. (4). By setting the gradient of the objective to zeros, we can get an analytic solution, i.e.,

$$\frac{\partial \mathcal{L}\left(\boldsymbol{\Delta}^{t+1}, \boldsymbol{c}^{t+1}, \boldsymbol{\mu}\right)}{\partial \mu_k} = 0 \Longrightarrow \mu_k^{t+1} = c_k^{t+1}, \quad k = 1, 2, \ldots, K. \tag{7}$$

Although all the update solutions are formulated in terms of full batch update, it can be easily extended to stochastic update using mini-batch data. To be specific, the update solution (Eq. (5)) of the network parameters $\boldsymbol{\Delta}$ naturally accepts mini-batch update for stochastic gradient optimization. Meanwhile, $\boldsymbol{c}$ are gradually updated following EMA. Previous literature demonstrates that EMA possesses good stability and efficiency in the scenario of the mini-batch update [39]. Therefore, our PC-X can deal with large-scale clustering problems.

### 3.3.2. Convergence analysis

Our decomposition-coordination optimization method falls into the scope of the batch block optimization framework [40]. In the following, we give the definition of Lipschitz smoothness, strong convexity and bounded gradients, which are standard for convex stochastic optimization [41, 42].

**Definition 1** ($L$-**Lipschitz smooth**) *A differentiable function* $f : \mathbb{R}^n \to \mathbb{R}$ *is said to be L-smooth if* $\forall x, y \in \mathbb{R}^n$, *we have* $\|\nabla f(x) - \nabla f(y)\| \leq L\|x - y\|$.

**Definition 2** ($u$-**Strong convexity**) *A function* $f : \mathbb{R}^n \to \mathbb{R}$ *is said to be is u-strongly convex for some* $u > 0$, *which means that* $\forall x, y \in \mathbb{R}^n$, *we have* $f(x) \geq f(y) + \langle \nabla f(y), y - x \rangle + \frac{u}{2}\|y - x\|^2$.

**Definition 3** (**Bounded gradients**) *If the gradients* $\nabla f(x)$ *of the objective function* $f(x)$ *are upper bounded by* $G$, *it means that* $\|\nabla f(x)\|^2 \leq G, \forall x \in \mathbb{R}^n$.

**Lemma 1** (**Quadratic upper bounds [43]**) *If the differentiable objective function* $\mathcal{L}(\Delta, c, \mu)$ *in Eq. (4) is partial L-Lipschitz smooth to the network parameters* $\Delta$, *i.e.,* $\forall \Delta_1, \Delta_2 \in \mathbb{R}^{|\Delta|}$, $\left\| \frac{\partial \mathcal{L}(\Delta, c, \mu)}{\partial \Delta}\big|_{\Delta=\Delta_1} - \frac{\partial \mathcal{L}(\Delta, c, \mu)}{\partial \Delta}\big|_{\Delta=\Delta_2} \right\| \leq L \|\Delta_1 - \Delta_2\|$. *Then, we can derive the following quadratic upper bounds:* $\mathcal{L}(\Delta_1, c, \mu) \leq \mathcal{L}(\Delta_2, c, \mu) + \left\langle \frac{\partial \mathcal{L}(\Delta, c, \mu)}{\partial \Delta}\big|_{\Delta=\Delta_2}, \Delta_1 - \Delta_2 \right\rangle + \frac{L}{2}\|\Delta_1 - \Delta_2\|^2$. ∎

**Lemma 2** (**Convexity upper bounds**) *If the objective function* $\mathcal{L}(\Delta, c, \mu)$ *in Eq. (4) satisfies the partially u-strongly convex w.r.t. the network parameter* $\Delta$ *for some* $u > 0$, *i.e.,* $\forall \Delta_1, \Delta_2 \in \mathbb{R}^{|\Delta|}$, $\mathcal{L}(\Delta_2, c, \mu) \geq \mathcal{L}(\Delta_1, c, \mu) + \left\langle \frac{\partial \mathcal{L}(\Delta, c, \mu)}{\partial \Delta}\big|_{\Delta=\Delta_1}, \Delta_1 - \Delta_2 \right\rangle + \frac{u}{2}\left\|\Delta_1 - \Delta_2\right\|^2$. *Then, we have:* $\mathcal{L}(\Delta_1, c, \mu) - \mathcal{L}(\Delta_2, c, \mu) \leq \frac{1}{2u}\left\| \frac{\partial \mathcal{L}(\Delta, c, \mu)}{\partial \Delta}\big|_{\Delta=\Delta_1} \right\|^2$, *which simply applies the optimal value of the quadratic function.* ∎

It is too strong to assume the objective function of a deep neural network is Lipschitz continuity and strongly convex w.r.t. the network parameter $\Delta$, while it is reasonable to assume it is Lipschitz continuity and strongly convex around a neighborhood of a local optima. Meanwhile, the two sup-problems w.r.t. $c, \mu$ are global convex with closed form solutions (See Eq. (6) and Eq. (7)).

**Theorem 1** (**Linear Convergence**) *Assume the objective* $\mathcal{L}(\Delta, c, \mu)$ *in Eq. (4) satisfies the assumptions of partial Lipschitz smooth, u-strong convexity and bounded gradients w.r.t. the network parameter* $\Delta$ *around a neighborhood of* $(\Delta^*, c^*, \mu^*)$, *which is a local optimal solution of the objective function* $\mathcal{L}(\Delta, c, \mu)$ *in expectation. Our algorithm converges linearly to optimal value* $(\Delta^*, c^*, \mu^*)$ *with a step-size* $\gamma$, *i.e.,* $\mathbb{E}[\mathcal{L}(\Delta^t, c^t, \mu^t) - \mathcal{L}(\Delta^*, c^*, \mu^*)] \leq (1 - 2u\gamma)^t \mathbb{E}[\mathcal{L}(\Delta^0, c^0, \mu^0) - \mathcal{L}(\Delta^*, c^*, \mu^*)] + \frac{\gamma LG}{4u}$, *where* $(\Delta^t, c^t, \mu^t)$ *is the solution at t-th iteration. The expectation is taken w.r.t. the stochastic mini-batch data.* ∎

According to our analysis in Theorem 1, as long as we set an appropriate learning rate $\gamma$ where $0 < \gamma < \frac{1}{2u}$, the objective function Eq. (4) will converge linearly.

### 3.3.3. Complexity analysis

We analyze the time and space complexity of our PC-X and popular AE/DGM-based deep clustering methods, i.e., VaDE, IDEC and DEPICT, in Table 2. Let $M$, $N$, $B$, $K$ and $d$ denote the number of epochs, the whole sample size, the mini-batch size, the cluster number and latent feature dimension, respectively. Assuming all of them adopt the same AE network architecture, we only analyze the extra time and space cost incurred by the clustering module.

Table 2: Algorithm complexity of the clustering module

| Baseline | VaDE[23] | IDEC[20] | DEPICT[18] | PC-X |
|---|---|---|---|---|
| Time | $\mathcal{O}(MNKd^2)$ | $\mathcal{O}(MNKd)$ | $\mathcal{O}(MNKd)$ | $\mathcal{O}(MNKd)$ |
| Space | $\mathcal{O}((B+2K)d + K)$ | $\mathcal{O}((N+K)d)$ | $\mathcal{O}((N+K)d)$ | $\mathcal{O}((B+K)d)$ |

From Table 2, we can find that: (1) since IDEC, DEPICT and our PC-X perform $k$-means clustering, they have the same extra time complexity. (2) Both IDEC and DEPICT are based on DEC [16] but

differ in their AE structure. Therefore, their clustering modules incur the same algorithm complexity. Compared to our PC-X using mini-batch ($B$) samples, IDEC and DEPICT are not memory efficient, since both of them need to repeatedly update the target probability with full batch ($N \gg B$) samples. (3) VaDE performs Gaussian Mixture Models (GMM) clustering with a diagonal variance matrix, resulting in a larger time complexity than our PC-X. It also updates the clustering module with mini-batch samples, but has an additional space cost $\mathcal{O}(Kd + K)$ for the diagonal variance matrix and the prior probability.

# 4. Empirical experiment

In this section, we conduct experiments on seven benchmark datasets to demonstrate the superiority of our PC-X in terms of clustering accuracy, universality, efficiency, and interpretability, respectively. The statistics of datasets are introduced in Table 3.

Table 3: The statistics of datasets

| Dataset | Type | #sample | #cluster | #dim | major | minor |
|---|---|---|---|---|---|---|
| MNIST [44] | Image | 70,000 | 10 | $28 \times 28$ | 11.3% | 9% |
| Fashion [45] | Image | 70,000 | 10 | $28 \times 28$ | 10% | 10% |
| YTF [46] | Image | 10,000 | 41 | $3 \times 55 \times 55$ | 6.9% | 0.3% |
| CIFAR-10 [47] | Image | 60,000 | 10 | $3 \times 32 \times 32$ | 10% | 10% |
| ImageNet-10 [48] | Image | 13,000 | 10 | $3 \times 24 \times 24$ | 10% | 10% |
| Reuters10K [16] | Text | 10,000 | 4 | 2,000 | 40.2% | 9.0% |
| HAR [49] | Signal | 10,299 | 6 | 561 | 18.9% | 13.7% |

For the imbalanced datasets YTF and Reteurs10K, we apply a re-weighted clustering loss (See A.3).

We compare our PC-X[2] with i) two classical clustering approaches: $k$-means [50] and GMM [51]; ii) two pretrained model based deep clustering methods: AE [10] and DEC [16]; iii) three AE/DGM based deep clustering methods: VaDE [23], IDEC [20] and DEPICT [18]; iv) six SSL-based deep clustering methods: IIC [5], PICA [14], CC [34], SPICE[3] [15], TCC [25] and DivClust [26]. Inspired by SSL-based clustering methods [15, 31], we use MoCo [52] to extract features for CIFAR-10 and ImageNet-10 considering their complexity, and evaluated all non-SSL-based baselines (including our PC-X) on the extracted features. Like SSL-based baselines, these feature extractors do not utilize any supervision regarding the datasets. For all methods, we set the number of clusters to the ground truth categories and evaluate performance with three clustering metrics: Accuracy (ACC), Normalized Mutual Information (NMI) and Adjusted Rand Index (ARI).

## 4.1. Comparison in terms of clustering performance and universality

Table 4: Comparisons of PC-X with standard clustering, pretrained model based deep clustering and AE/DGM-based deep clustering methods. Best marked in bold, second best underlined.

| Baselines | MNIST | | | Fashion | | | YTF | | | CIFAR-10 | | | ImageNet-10 | | | Reuters10K | | | HAR | | |
|---|---|---|---|---|---|---|---|---|---|---|---|---|---|---|---|---|---|---|---|---|---|
| | ACC | NMI | ARI | ACC | NMI | ARI | ACC | NMI | ARI | ACC | NMI | ARI | ACC | NMI | ARI | ACC | NMI | ARI | ACC | NMI | ARI |
| $k$-means | 53.5 | 50.0 | 36.6 | 47.6 | 51.2 | 34.9 | 58.3 | 76.9 | 53.8 | 33.5 | 41.0 | 9.8 | 88.3 | 79.1 | 76.6 | 51.7 | 31.1 | 20.8 | 60.0 | 58.9 | 40.1 |
| GMM | 43.4 | 37.1 | 23.7 | 49.8 | 53.3 | 34.9 | 59.4 | 77.9 | 55.1 | 78.5 | 74.4 | 68.2 | 88.5 | 79.2 | 76.9 | 73.6 | 48.5 | 46.5 | 59.7 | 59.4 | 47.1 |
| AE | 86.2 | 77.4 | 77.4 | 54.1 | 56.1 | 39.5 | 58.1 | 76.5 | 53.7 | 83.4 | 74.6 | 69.0 | 84.7 | 75.1 | 71.7 | 79.1 | 53.9 | 60.0 | 68.9 | 70.4 | 62.0 |
| DEC | 88.2 | 80.9 | 78.4 | 57.1 | 58.8 | 43.5 | 59.9 | 80.3 | 59.3 | 85.1 | 76.6 | 72.5 | 87.1 | 78.7 | 75.8 | 73.0 | 47.3 | 49.4 | 68.6 | 67.0 | 59.1 |
| IDEC | 89.8 | 84.3 | 81.7 | 52.9 | 56.8 | 42.9 | 65.1 | 78.2 | 55.7 | 84.8 | 76.5 | 72.4 | 87.2 | 78.7 | 76.0 | 72.9 | 47.2 | 49.0 | 68.7 | 74.0 | 64.2 |
| VaDE | 94.5 | 87.6 | 88.2 | 45.1 | 55.4 | 36.6 | 31.0 | 50.6 | 24.1 | N.A. | N.A. | N.A. | N.A. | N.A. | N.A. | 79.9 | 51.2 | 58.0 | 84.3 | 75.3 | 69.9 |
| DEPICT | 95.3 | 90.4 | 90.1 | 55.7 | 57.9 | 42.0 | 61.1 | 79.2 | 57.6 | 84.0 | 75.3 | 70.2 | 85.5 | 76.2 | 73.1 | 80.0 | 56.1 | 62.0 | 69.1 | 70.8 | 62.3 |
| PC-X | 97.5 | 93.5 | 94.7 | 64.1 | 67.0 | 51.6 | 66.7 | 82.8 | 62.8 | 87.9 | 79.8 | 77.0 | 92.3 | 83.4 | 84.0 | 82.2 | 57.0 | 63.1 | 87.1 | 80.1 | 75.1 |

We demonstrate the superiority of PC-X on datasets with various data types. As SSL-based methods were proposed for complex image datasets, we applied them only on ImageNet-10 and CIFAR-10.

Table 4 shows that: (1) PC-X consistently achieves superiority clustering accuracy than AE/DGM-based baselines on various datasets, this is because our CI-Unit can suppress sample-specific details,

---

[2]For all baselines, results were retrieved from the literature or computed by us when not found. Algorithms with missing values are because the original paper did not report and it was difficult to get a satisfying score.

[3]SPICE is a three-stage algorithm, we only compare with SPICE at the second stage, as the last fine-tuning stage is generally applicable to all deep clustering methods.

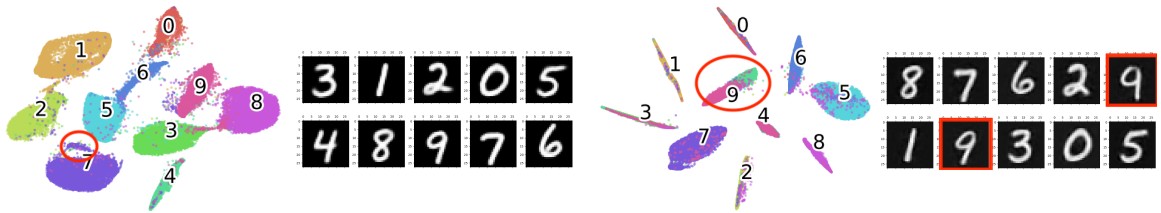

| (a) t-SNE of PC-X | (b) centroids of PC-X | (c) t-SNE of IDEC | (d) centroids of IDEC |

Figure 2: t-SNE for latent embedding (**1st row**) and centroids reconstruction (**2nd row**) on MNIST.

thereby contributing to better clustering-friendly representation. (2) AE/DGM-based deep clustering methods behave differently on various types of datasets, comparable or even inferior to AE on Fashion and Reuters10K, due to their failure to reconcile both goals of self-reconstruction and clustering.

Table 5 shows that PC-X can achieve higher or comparable clustering accuracy over SSL-based deep clustering methods on complex image datasets through a better feature extractor. It ascribes the success of SOTA SSL-based deep clustering methods to a better feature extractor instead of their respectively complex but less interpretable clustering modules. Back to universality, our PC-X is more advantageous than SSL-based clustering baselines due to its ability in handling non-image datasets, e.g., Reuters10K (text) and HAR (signal).

Table 5: Comparisons of PC-X with SSL-based deep clustering methods. Best marked in bold, second best underlined.

| Method | CIFAR-10 | | | ImageNet-10 | | |
|---|---|---|---|---|---|---|
| | ACC | NMI | ARI | ACC | NMI | ARI |
| IIC | 61.7 | 51.3 | 41.1 | 24.7 | 15.4 | 10.0 |
| PICA | 64.5 | 56.1 | 46.7 | 85.0 | 78.2 | 73.3 |
| CC | 79.0 | 70.5 | 63.7 | 89.3 | $\underline{85.9}$ | 82.2 |
| SPICE$_s$ | 83.8 | 73.4 | 70.5 | $\underline{92.1}$ | 82.8 | 83.6 |
| TCC | **90.6** | $\underline{79.0}$ | $\underline{73.3}$ | 89.7 | 84.8 | 82.5 |
| DivClust | 81.9 | 72.4 | 68.1 | 91.8 | **87.9** | **85.1** |
| PC-X | $\underline{87.9}$ | **79.8** | **77.0** | **92.4** | 83.4 | $\underline{84.0}$ |

## 4.2. Visualization analyses for the interpretability

We visualize the latent embedding and the corresponding centroids on MNIST to showcase the interpretability of PC-X in Fig. 2. Additionally, we compare our visual results with IDEC, whose model is almost the same as our PC-X but lacks the CI-Unit.

It is notable that the centroids reconstruction of our PC-X are exactly the realistic images (Fig. 2b), which indicates PC-X captures legible exemplars and performs semantic clustering in the latent space. Regarding IDEC (Fig. 2c, 2d), the embedding of some categories are mixed, causing redundancy centroid reconstructions, i.e., digit 9, and incomplete identified categories, i.e., digit 4. This is because its self-reconstruction loss, which doesn't take clustering centroids into account, contains too many sample-specific details. The clustering results are distracted by similar digits, i.e., 4 and 9. This observation also emphasizes the importance of interpretability. Well-separated embedding in the latent space (Fig. 2c) does not guarantee semantically meaningful clustering results (Fig. 2d).

## 4.3. Efficiency: fast and stable convergence with stochastic optimization

In Fig. 3, we collect the clustering accuracy of PC-X on mini-batch (PC-X-mini)/ full-batch (PC-X) of MNIST/Fashion as well as the clustering accuracy of other popular clustering methods [53], e.g., $k$-means, GMM and hierarchical agglomerative clustering (AGG) [54], respectively.

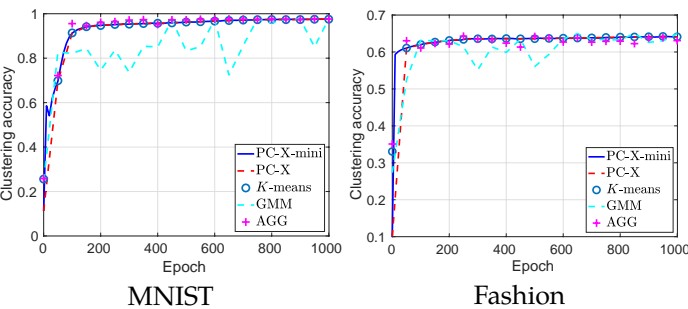

MNIST         Fashion

Figure 3: Clustering with fast and stable convergence.

Fig. 3 verifies that: (1) PC-X converges fast ($epoch < 100$) on both two datasets and continue to improve gradually later, which benefits from our efficient joint optimization scheme. (2) The clustering accuracy of PC-X on mini-batch (PC-X-mini) keeps align with that of full-batch (PC-X) and $k$-means during the whole clustering process, which demonstrates the superiority of our slow exemplars cal-

ibration strategy. (3) PC-X delivers clustering-friendly (i.e., well-separated) feature representations since PC-X achieves a similar clustering accuracy as classical shallow clustering methods, especially hierarchical clustering method AGG. Meanwhile, it justifies our choice that simple $k$-means is sufficient for clustering in a clustering-friendly latent space we deliver.

To further demonstrate the efficiency of our PC-X on the large-scale dataset, we evaluate it on the entire ImageNet dataset with 1,000 clusters and 1,281,167 samples. Similarly, we apply MoCo [52] to extract features for ImageNet before performing PC-X. The clustering results are ACC: 38.4, NMI: 68.8, ARI: 31.7. These are the first results on ImageNet reported for deep AE-based clustering.

### 4.4. Ablation study on network components

We compare different variants of PC-X with most related baselines, i.e., IDEC and AE, to analyze the efficacy of different components in PC-X. Different strategies for updating the centroids are studied. "PC-X-no CI" denotes PC-X without the CI-Unit; "PC-X-Fix" denotes the centroids are fixed after initialization.

Table. 6 summarizes the clustering accuracy (i.e., ACC and NMI) of PC-X and its variants. It shows that (1) PC-X with all components can achieves the best results over other baselines, which demonstrates the efficacy of our CI-Unit and joint optimization strategy. (2) Without a proper strategy, two variants of PC-X, i.e., "PC-X-no CI" and "PC-X-Fix", drops significantly, since they fail to reconcile the conflicts between the goal of self-reconstruction and clustering, or the optimization between the network parameters and the clustering centroids, respectively.

Table 6: Ablation study on network components using MNIST and Fashion datasets. Clustering performance (ACC&NMI) of PC-X and its variants using the model output or $k$-means, respectively. Best results marked in bold.

| Ablation | | PC-X | PC-X-no CI | PC-X-Fix | IDEC | AE |
|---|---|---|---|---|---|---|
| MNIST | ACC | **97.5** | 94.1 | 57.6 | 89.8 | 86.2 |
| | NMI | **93.5** | 87.8 | 76.8 | 84.3 | 77.4 |
| Fashion | ACC | **66.7** | 57.4 | 50.7 | 52.9 | 54.1 |
| | NMI | **64.3** | 62.0 | 61.7 | 56.8 | 56.1 |

### 4.5. Effectiveness of various losses

Fig. 4 visualizes the reconstruction loss, the clustering loss and the average group assignment $\frac{1}{N}\sum_n \max_k \lambda_{nk}$. It shows that: (1) the average group assignment gradually decreases after initialization in pursuit of a good feature representation. (2) After a burn-in period (#epoch $\leq 250$), both two losses remain stable, and the average group assignment levels off. (3) After we activate ME regularization, the average group assignment keeps rising to nearly one; while the reconstruction loss remains stable and the clustering loss slightly increases. It means the ME regularization helps to distill a well-separate latent structure and improve the clustering stability.

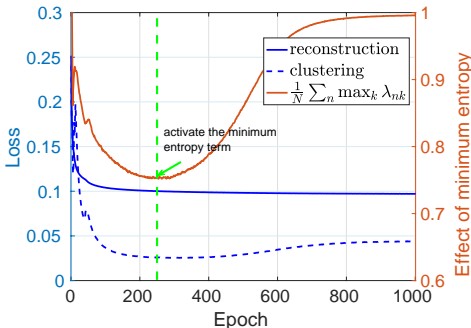

Figure 4: Effectiveness of various losses.

## 5. Conclusion

This paper presents profound clustering via slow exemplar (PC-X), which casts an interpretable clustering framework for large-scale applications. It fosters a set of legible exemplars for clustering in the latent space without accessing the full-batch data concurrently. Since PC-X conducts clustering based on the similarity in the latent space, the clustering performance is affected by the choice of the AE network. Extra constraints [55, 56] can be introduced to facilitate a semantics-consistent latent embedding, so as to reduce reliance on network structure. Meanwhile, PC-X's reliance on non-convex $k$-means may impact its performance. Investigating the integration of other differential clustering losses could yield a more robust clustering methodology. Further, our PC-X paradigm inspires a stable and efficient way of Bayesian inference to stitch deep networks with complex Bayesian models. In the future, we will extend PC-X for non-parametric Bayes and hierarchical models.

# Acknowledgements

YP, YY and IT were supported in part by the A*STAR Centre for Frontier AI Research. YP and YY were also supported in part by the A*STAR Career Development Fund No.C222812019.

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

## A. Appendix

### A.1. Network structure of our profound Clustering (PC-X)

Fig. 5 shows the whole structure of our Profound Clustering (PC-X), which encodes the clustering module into the auto-encoder network, which is an inherently interpretable clustering framework and universally applicable for complex data.

### A.2. PC-X-Div for diversity promotion

It was claimed that adding diversification constraints to encourage the diversity among clustering centroids, which was found to yield better clustering performance (e.g., [35, 36]). However, such repulsive regularization terms are inherently highly non-convex, adding them into the objective

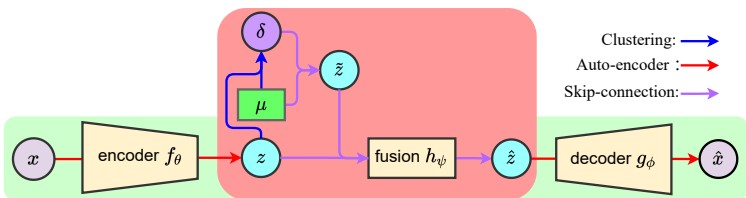

Figure 5: Profound Clustering (PC-X): encoding clustering into deep auto-encoder network

function would cause additional optimization difficulty [57]. Inspired by these, we add a minimum entropy (ME) regularization over the soft group assignment $\lambda_{nk}$ as follows:

$$\mathcal{L}_{ME}(\boldsymbol{\Delta}) = -\frac{1}{N} \sum_{n=1}^{N} \sum_{k=1}^{K} \lambda_{nk} \log \lambda_{nk}, \tag{8}$$

where $\{\lambda_{nk}\}_{n=1,k=1}^{N,M}$ is the intermediate variable calculated via Eq. (2). Note Eq. (8) is optimized only for the network parameters to avoid instability which will not cause extra optimization difficulty.

Ideally, when Eq. (8) achieves optima, the soft assignment should be nearly the same as the corresponding hard assignment, i.e., $\lambda_{nk} \to \delta_{nk}$. It means that the distance of a sample to its closest centroid should be much smaller than other centroids, denoting the latent embedding is well-separated as desired. This method shares a similar spirit with the sharpening technique [58] used in DEC [16] and its derivatives [18, 20], but minimum entropy is calculated at the sample level which is superior in large-scale applications using mini-batch update.

Therefore, we summarize the objective of our PC-X with skip-connection as follows:

$$\mathcal{L}_{PC\text{-}X\text{-}Div}(\boldsymbol{\Delta}, \boldsymbol{\mu}) = \underbrace{\frac{1}{N} \sum_{n=1}^{N} D(x_n \| g_\phi(h_\psi(z_n, \tilde{z}_n)))}_{Reconstruction\ loss} + \eta_1 \underbrace{\frac{1}{N} \sum_{n=1}^{N} \sum_{k=1}^{K} \delta_{nk} \| z_n - \mu_k \|_2^2}_{Clustering\ loss} + \eta_2 \underbrace{\left( -\frac{1}{N} \sum_{n=1}^{N} \sum_{k=1}^{K} \lambda_{nk} \log \lambda_{nk} \right)}_{Minimum\ Entropy},$$

$$\tag{9}$$

where $z_n = f_\theta(x_n)$. $\tilde{z}_n$ denotes the closest clustering centroid $\mu_{\{\text{argmax}_j \lambda_{nj}\}}$ of the latent embedding $z_n$ according to Eq. (3). $\boldsymbol{\Delta} = \{\theta, \varphi, \phi\}$ stands for network parameters for both auto-encoder and skip-connection. $\eta_1, \eta_2$ are the trade-off parameters.

It is noted that all three losses are calculated per sample, implying that PC-X-Div supports stochastic optimization using mini-batch data as well.

### A.3. PC-X-IMBA for imbalanced data clustering

Vanilla clustering loss does not take into consideration the scenario of imbalanced clustering. Thus, it would suffer from inferior clustering performance when clustering size is imbalanced. Because the major clustering will dominate the clustering loss, causing the minor cluster to be less explored.

Let $\boldsymbol{\alpha} = [\alpha_1, \alpha_2, \ldots, \alpha_K]$ denote the clustering size, and we define

$$\omega_k = \frac{\alpha_k K}{\sum_{j=1}^{K} \alpha_j} = \frac{\alpha_k}{\bar{\alpha}}, \quad k = 1, 2, \ldots, K, \tag{10}$$

where $\bar{\alpha}$ denotes the average clustering size.

Then, we rearrange the clustering loss as follows,

$$
\mathcal{L}_{cluster}(\boldsymbol{\mu}) = \frac{1}{N}\sum_{n=1}^{N}\sum_{k=1}^{K}\delta_{nk}\|z_n-\mu_k\|_2^2 \overset{①}{=} \sum_{k=1}^{K}\left[\frac{1}{N}\sum_{n=1}^{N}\delta_{nk}\|z_n-\mu_k\|_2^2\right] \overset{②}{=} \sum_{k=1}^{K}\left[\frac{1}{N}\sum_{x\in S_k}\|f_\theta(x)-\mu_k\|_2^2\right]
$$

$$
\overset{③}{=} \frac{1}{K}\sum_{k=1}^{K}\boxed{\frac{K\alpha_k}{N}}\left[\frac{1}{\alpha_k}\sum_{x\in S_k}\|f_\theta(x)-\mu_k\|_2^2\right] \overset{④}{=} \frac{1}{K}\sum_{k=1}^{K}\omega_k\underbrace{\left[\frac{1}{\alpha_k}\sum_{x\in S_k}\|f_\theta(x)-\mu_k\|_2^2\right]}_{Inter\text{-}cluster\ Variance}, \qquad (11)
$$

where ① is valid by swapping the order of the two summarization operators. ② is because the existence of the hard assignment $\delta_{nk}\in\{0,1\}$ where the summarization is actually taken over the samples assigned the cluster $S_k$. ③ is because $\frac{1}{N}=\frac{1}{K}\times\frac{K\alpha_k}{N}\times\frac{1}{\alpha_k}$. ④ is according to Eq. (10).

The vanilla clustering loss (Eq. (11)) is actually a weighted summarization of inter-cluster variance, which places a higher weight for a larger cluster, i.e., $\frac{K\alpha_k}{N}=\frac{\alpha_k}{\bar{\alpha}}>1$, and vice versa. $\bar{\alpha}$ denotes the average clustering size. We argue that this term can be removed, especially for clustering with imbalanced data. Particularly, the new clustering loss will no longer bias towards major clusters and ensure good clustering performance in imbalanced data clustering.

**Reweighting with inverse clustering size.** The cluster assignment is exactly what we target for, which unfortunately is not available before training. Although it can be estimated during clustering, it is never the optima before converging. Since it cannot be removed directly, we consider weakening the effect of this term in the clustering loss. Compared with the (sample-level) cluster assignment, the (cluster-level) clustering size can be estimated more reliable and accurate during the clustering process, we therefore reweight the clustering loss with inverse clustering size to lend an ear to small clusters. To be specific,

$$
\mathcal{L}_{reweight\text{-}cluster}(\boldsymbol{\mu}) = \frac{1}{N}\sum_{n=1}^{N}\boxed{\frac{1}{\omega_k}}\sum_{k=1}^{K}\delta_{nk}\|z_n-\mu_k\|_2^2 \overset{①}{=} \frac{1}{K}\sum_{k=1}^{K}\underbrace{\left[\frac{1}{\alpha_k}\sum_{x\in S_k}\|f_\theta(x)-\mu_k\|_2^2\right]}_{Inter\text{-}cluster\ Variance},
$$

where ① is derived following Eq. (11).

To sum up, we introduce the PC-X-IMBA tailor-designed for clustering imbalanced data based the cluster-lever reweight in the following,

$$
\mathcal{L}_{PC\text{-}X\text{-}IMBA}(\boldsymbol{\Delta},\boldsymbol{\mu}) = \mathcal{L}_{AE}(\boldsymbol{\Delta}) + \eta_1\mathcal{L}_{reweight\text{-}cluster}(\boldsymbol{\mu}) + \eta_2\mathcal{L}_{ME}(\boldsymbol{\lambda}). \qquad (12)
$$

Compared with PC-X, PC-X-IMBA is superior for clustering imbalanceddata because

- In terms of balance data, i.e., $\alpha_1\approx\ldots\approx\alpha_K\approx\bar{\alpha}$, we have $\omega_k\approx 1$. PC-X-IMBA degenerates to PC-X. Meanwhile, for an average cluster in the imbalanced dataset, i.e., $\alpha_k\approx\bar{\alpha}$ and $\omega_k\approx 1$ which means no extra action required.

- In terms of a minority cluster $\alpha_k<\bar{\alpha}$ in the imbalance dataset, we have $\frac{1}{\omega_k}=\frac{\bar{\alpha}}{\alpha_k}>1$ which gives a higher weight to the sample assigned to a minority cluster.

- In terms of a major cluster $\alpha_k>\bar{\alpha}$ in the imbalance dataset, we have $\frac{1}{\omega_k}=\frac{\bar{\alpha}}{\alpha_k}<1$ which gives a lower weight to the sample assigned to a major cluster.

### A.3.1. Estimating the clustering size via EMA

Inspired by the EMA estimation for the clustering centroids, we suggest estimating the cluster size $\boldsymbol{\alpha}=[\alpha_1,\alpha_2,\ldots,\alpha_K]$ on the fly as well. Similarly, $\boldsymbol{\alpha}$ is online calibrated at each iteration as follow:

$$
\alpha_k^{t+1} = \kappa\alpha_k^t + (1-\kappa)\sum_{n=1}^{N}\delta_{nk}, \quad k=1,2,\ldots,K. \qquad (13)
$$

**Algorithm 1** Decomposition-coordination optimization

---

1: **Input:** dataset $X = \{x_n\}_{n=1}^N$, AE network $\{f_\theta, g_\phi, h_\psi\}$, #centroids $K$, #iterations $M$, batch size $B$.
2: **Output:** encoder $f_\theta$, centroids $\boldsymbol{\mu}$, group assignment $\mathcal{S}$.
3: **Initialize:** network parameters $\boldsymbol{\Delta}$ and centroids $\boldsymbol{\mu}$.
4: **for** iteration $= 1, 2, \ldots, M$ **do**
5:    **for** mini-batch $= 1, 2, \ldots, \lceil \frac{N}{B} \rceil$ **do**
6:       sample a mini-batch from $X$;
7:       update group assignment $\{\delta_{nk}\}$ by Eq. (2);
8:       update network parameters $\boldsymbol{\Delta}$ by Eq. (5);
9:       update intermediate clustering status $\boldsymbol{c}$ by Eq. (6);
10:      update clustering centroids $\boldsymbol{\mu}$ by Eq. (7).
11:    **end for**
12: **end for**

---

To ensure stability, we further adopt Laplace smoothing to smooth clustering size. Namely,

$$\alpha_k = \frac{(\alpha_k + \epsilon)(\sum_{j=1}^K \alpha_j)}{\sum_{j=1}^K \alpha_j + \epsilon K}, \quad k = 1, 2, \ldots, K, \tag{14}$$

where the pseudo count $\epsilon > 0$ is the smoothing parameter. In the real experiment, $\epsilon$ is set to a small value ($\approx 1 \times 10^{-5}$) for a balance clustering scenario and a large value ($\approx 5$) for an imbalanced data clustering scenario.

A burn-in period is introduced for the minimum entropy regularization (Eq. (8)) and imbalanced data clustering PC-X-IMBA (Eq. (12)) since the estimation is less informative initially. Namely, during the initial period (#epoch $\leq 250$) of the training epochs, $\eta_2 = 0$ and $\omega_k = 1$ when the clustering module is not properly learned.

## A.4. Detailed derivation for Eq.(6)

By setting the gradient of the objective to zeros, we can get an analytic solution.

$$\frac{\partial \mathcal{L}\left(\boldsymbol{\Delta}^{t+1}, \boldsymbol{c}, \boldsymbol{\mu}^t\right)}{\partial c_k} = 0, \quad k = 1, 2, \ldots, K.$$

$$\Longrightarrow 2\eta_1 \frac{1}{N} \sum_{n=1}^N \delta_{nk}(c_k - z_n) + 2\rho(c_k - \mu_k^t) = 0$$

$$\Longrightarrow (\frac{\eta_1}{N} \sum_{n=1}^N \delta_{nk} + \rho)c_k = \rho \mu_k^t + \frac{\eta_1}{N} \sum_{n=1}^N \delta_{nk} z_n. \tag{15}$$

$$\Longrightarrow c_k = \frac{\rho \mu_k^t + \frac{\eta_1}{N} \sum_{n=1}^N \delta_{nk} z_n}{\rho + \frac{\eta_1}{N} \sum_{n=1}^N \delta_{nk}},$$

where $k = 1, 2, \ldots, K$.

Therefore, we let

$$c_k^{t+1} = \frac{\rho \mu_k^t + \frac{\eta_1}{N} \sum_{n=1}^N \delta_{nk} z_n}{\rho + \frac{\eta_1}{N} \sum_{n=1}^N \delta_{nk}}, \quad k = 1, 2, \ldots, K. \tag{16}$$

## A.5. Proof for Theorem 1

**Theorem 1** [*Linear Convergence*] *Assume the objective function $\mathcal{L}(\boldsymbol{\Delta}, \boldsymbol{c}, \boldsymbol{\mu})$ in Eq. (4) satisfies the assumptions of partial Lipschitz smooth, u-strong convexity and bounded gradients w.r.t. the network parameter $\boldsymbol{\Delta}$ around a neighborhood of $(\boldsymbol{\Delta}^*, \boldsymbol{c}^*, \boldsymbol{\mu}^*)$, which is a local optimal solution of the objective function $\mathcal{L}(\boldsymbol{\Delta}, \boldsymbol{c}, \boldsymbol{\mu})$*

*in expectation. Our algorithm 1 converges linearly to optimal value $(\boldsymbol{\Delta}^*, \boldsymbol{c}^*, \boldsymbol{\mu}^*)$ with a constant step-size $\gamma$, as given below,*

$$\mathbb{E}\left[\mathcal{L}(\boldsymbol{\Delta}^t, \boldsymbol{c}^t, \boldsymbol{\mu}^t) - \mathcal{L}(\boldsymbol{\Delta}^*, \boldsymbol{c}^*, \boldsymbol{\mu}^*)\right] \leq (1 - 2u\gamma)^t \mathbb{E}\left[\mathcal{L}(\boldsymbol{\Delta}^0, \boldsymbol{c}^0, \boldsymbol{\mu}^0) - \mathcal{L}(\boldsymbol{\Delta}^*, \boldsymbol{c}^*, \boldsymbol{\mu}^*)\right] + \frac{\gamma L G}{4u},$$

*where $(\boldsymbol{\Delta}^t, \boldsymbol{c}^t, \boldsymbol{\mu}^t)$ is the solution at $t$-th iteration over mini-batch $B^t$. The expectation is taken with regard to the stochastic mini-batch data.*

**Proof:** Give a mini-batch dataset $B^t$, according to our decomposition-coordination optimization, we have that

$$\mathcal{L}(\boldsymbol{\Delta}^{t+1}, \boldsymbol{c}^{t+1}, \boldsymbol{\mu}^{t+1}) \overset{\textcircled{1}}{\leq} \mathcal{L}(\boldsymbol{\Delta}^{t+1}, \boldsymbol{c}^{t+1}, \boldsymbol{\mu}^t) \overset{\textcircled{2}}{\leq} \mathcal{L}(\boldsymbol{\Delta}^{t+1}, \boldsymbol{c}^t, \boldsymbol{\mu}^t), \tag{17}$$

where $\textcircled{1}$ is valid according to the definition of subproblem in Eq. (7). $\textcircled{2}$ is because of the definition of subproblem Eq. (6).

Meanwhile, since $\boldsymbol{\Delta}^{t+1} = \boldsymbol{\Delta}^t - \gamma \frac{\partial \mathcal{L}(\boldsymbol{\Delta}, \boldsymbol{c}^t, \boldsymbol{\mu}^t)}{\partial \boldsymbol{\Delta}}\Big|^{B^t}_{\boldsymbol{\Delta}=\boldsymbol{\Delta}^t}$ [4] according to Eq. (5), we have

$$\mathcal{L}(\boldsymbol{\Delta}^{t+1}, \boldsymbol{c}^t, \boldsymbol{\mu}^t) - \mathcal{L}(\boldsymbol{\Delta}^t, \boldsymbol{c}^t, \boldsymbol{\mu}^t)$$

$$\overset{\textcircled{1}}{\leq} \left\langle \frac{\mathcal{L}(\boldsymbol{\Delta}, \boldsymbol{c}^t, \boldsymbol{\mu}^t)}{\partial \boldsymbol{\Delta}}\Big|_{\boldsymbol{\Delta}=\boldsymbol{\Delta}^t}, \boldsymbol{\Delta}^{t+1} - \boldsymbol{\Delta}^t \right\rangle + \frac{L}{2}\|\boldsymbol{\Delta}^{t+1} - \boldsymbol{\Delta}^t\|^2$$

$$\overset{\textcircled{2}}{=} -\gamma \left\langle \frac{\mathcal{L}(\boldsymbol{\Delta}, \boldsymbol{c}^t, \boldsymbol{\mu}^t)}{\partial \boldsymbol{\Delta}}\Big|_{\boldsymbol{\Delta}=\boldsymbol{\Delta}^t}, \frac{\partial \mathcal{L}(\boldsymbol{\Delta}, \boldsymbol{c}^t, \boldsymbol{\mu}^t)}{\partial \boldsymbol{\Delta}}\Big|^{B^t}_{\boldsymbol{\Delta}=\boldsymbol{\Delta}^t} \right\rangle + \frac{\gamma^2 L}{2}\left\|\frac{\partial \mathcal{L}(\boldsymbol{\Delta}, \boldsymbol{c}^t, \boldsymbol{\mu}^t)}{\partial \boldsymbol{\Delta}}\Big|^{B^t}_{\boldsymbol{\Delta}=\boldsymbol{\Delta}^t}\right\|^2$$

$$\overset{\textcircled{3}}{\leq} -\gamma \left\langle \frac{\mathcal{L}(\boldsymbol{\Delta}, \boldsymbol{c}^t, \boldsymbol{\mu}^t)}{\partial \boldsymbol{\Delta}}\Big|_{\boldsymbol{\Delta}=\boldsymbol{\Delta}^t}, \frac{\partial \mathcal{L}(\boldsymbol{\Delta}, \boldsymbol{c}^t, \boldsymbol{\mu}^t)}{\partial \boldsymbol{\Delta}}\Big|^{B^t}_{\boldsymbol{\Delta}=\boldsymbol{\Delta}^t} \right\rangle + \frac{\gamma^2 L G}{2}$$

where $\textcircled{1}$ follows Lemma 1, $\textcircled{2}$ is valid because the substitution using the stochastic gradient update. $\textcircled{3}$ applies the bounded gradient.

Subtracting optimal objective value $\mathcal{L}(\boldsymbol{\Delta}^*, \boldsymbol{c}^*, \boldsymbol{\mu}^*)$ and then taking expectation on both sides w.r.t. mini-batch $B^t$, we have,

$$\mathbb{E}\left[\mathcal{L}(\boldsymbol{\Delta}^{t+1}, \boldsymbol{c}^t, \boldsymbol{\mu}^t) - \mathcal{L}(\boldsymbol{\Delta}^*, \boldsymbol{c}^*, \boldsymbol{\mu}^*)\right] - \mathbb{E}\left[\mathcal{L}(\boldsymbol{\Delta}^t, \boldsymbol{c}^t, \boldsymbol{\mu}^t) - \mathcal{L}(\boldsymbol{\Delta}^*, \boldsymbol{c}^*, \boldsymbol{\mu}^*)\right]$$

$$\overset{\textcircled{1}}{\leq} -\gamma \left\langle \frac{\mathcal{L}(\boldsymbol{\Delta}, \boldsymbol{c}^t, \boldsymbol{\mu}^t)}{\partial \boldsymbol{\Delta}}\Big|_{\boldsymbol{\Delta}=\boldsymbol{\Delta}^t}, \mathbb{E}\left[\frac{\partial \mathcal{L}(\boldsymbol{\Delta}, \boldsymbol{c}^t, \boldsymbol{\mu}^t)}{\partial \boldsymbol{\Delta}}\Big|^{B^t}_{\boldsymbol{\Delta}=\boldsymbol{\Delta}^t}\right] \right\rangle + \frac{\gamma^2 L G}{2}$$

$$= -\gamma \mathbb{E}\left\|\frac{\partial \mathcal{L}(\boldsymbol{\Delta}, \boldsymbol{c}^t, \boldsymbol{\mu}^t)}{\partial \boldsymbol{\Delta}}\Big|_{\boldsymbol{\Delta}=\boldsymbol{\Delta}^t}\right\|^2 + \frac{\gamma^2 L G}{2}$$

$$\overset{\textcircled{2}}{\leq} -2u\gamma \mathbb{E}\left[\mathcal{L}(\boldsymbol{\Delta}^t, \boldsymbol{c}^t, \boldsymbol{\mu}^t) - \mathcal{L}(\boldsymbol{\Delta}^*, \boldsymbol{c}^*, \boldsymbol{\mu}^*)\right] + \frac{\gamma^2 L G}{2}$$

$$\implies \mathbb{E}\left[\mathcal{L}(\boldsymbol{\Delta}^{t+1}, \boldsymbol{c}^t, \boldsymbol{\mu}^t) - \mathcal{L}(\boldsymbol{\Delta}^*, \boldsymbol{c}^*, \boldsymbol{\mu}^*)\right] \leq (1 - 2u\gamma)\mathbb{E}\left[\mathcal{L}(\boldsymbol{\Delta}^t, \boldsymbol{c}^t, \boldsymbol{\mu}^t) - \mathcal{L}(\boldsymbol{\Delta}^*, \boldsymbol{c}^*, \boldsymbol{\mu}^*)\right] + \frac{\gamma^2 L G}{2}$$

$$\overset{\textcircled{3}}{\implies} \mathbb{E}\left[\mathcal{L}(\boldsymbol{\Delta}^{t+1}, \boldsymbol{c}^{t+1}, \boldsymbol{\mu}^{t+1}) - \mathcal{L}(\boldsymbol{\Delta}^*, \boldsymbol{c}^*, \boldsymbol{\mu}^*)\right] \leq (1 - 2u\gamma)\mathbb{E}\left[\mathcal{L}(\boldsymbol{\Delta}^t, \boldsymbol{c}^t, \boldsymbol{\mu}^t) - \mathcal{L}(\boldsymbol{\Delta}^*, \boldsymbol{c}^*, \boldsymbol{\mu}^*)\right] + \frac{\gamma^2 L G}{2}$$

where $\textcircled{1}$ is valid under the assumption of bounded gradients. $\textcircled{2}$ is derived following Lemma 2 under the assumption of the $u$-strong convexity. $\textcircled{3}$ follows the inequality in Eq. (17).

---

[4]The superscript $B^t$ denotes the stochastic gradient update over mini-batch $B^t$.

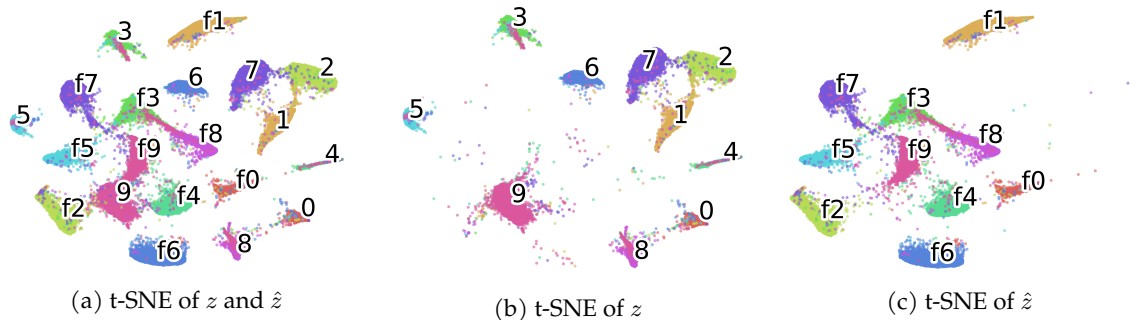

(a) t-SNE of $z$ and $\hat{z}$      (b) t-SNE of $z$      (c) t-SNE of $\hat{z}$

Figure 6: t-SNE of the feature representation before (i.e., $z$) and after (i.e., $\hat{z}$) the fusion layer on MNIST.

Applying this inequality recursively, we obtain

$$\mathbb{E}\left[\mathcal{L}(\boldsymbol{\Delta}^t, \boldsymbol{c}^t, \boldsymbol{\mu}^t) - \mathcal{L}(\boldsymbol{\Delta}^*, \boldsymbol{c}^*, \boldsymbol{\mu}^*)\right] \leq (1 - 2u\gamma)^t \mathbb{E}\left[\mathcal{L}(\boldsymbol{\Delta}^0, \boldsymbol{c}^0, \boldsymbol{\mu}^0) - \mathcal{L}(\boldsymbol{\Delta}^*, \boldsymbol{c}^*, \boldsymbol{\mu}^*)\right] + \sum_{i=1}^{t}(1 - 2u\gamma)^{(i-1)}\frac{\gamma^2 LG}{2}$$

$$\overset{\textcircled{1}}{\leq} (1 - 2u\gamma)^t \mathbb{E}\left[\mathcal{L}(\boldsymbol{\Delta}^0, \boldsymbol{c}^0, \boldsymbol{\mu}^0) - \mathcal{L}(\boldsymbol{\Delta}^*, \boldsymbol{c}^*, \boldsymbol{\mu}^*)\right] + \frac{1}{2u\gamma} \times \frac{\gamma^2 LG}{2}$$

$$= (1 - 2u\gamma)^t \mathbb{E}\left[\mathcal{L}(\boldsymbol{\Delta}^0, \boldsymbol{c}^0, \boldsymbol{\mu}^0) - \mathcal{L}(\boldsymbol{\Delta}^*, \boldsymbol{c}^*, \boldsymbol{\mu}^*)\right] + \frac{\gamma LG}{4u},$$

where $\textcircled{1}$ is valid because $\sum_{i=0}^{t} r^i \leq \sum_{i=0}^{\infty} r^i = \frac{1}{1-r}, \forall 0 < r < 1$. $\blacksquare$

## A.6. Functionality analysis of the "fully-connected layer" in CI-Unit

The fusion layer reconciles the goals of clustering and self-reconstruction, ensuring that the feature representation before fusion is suitable for clustering, while that after fusion is suitable for reconstruction. The following two experiments on MNIST verify the above claim:

1. We visualized the feature representation before (i.e., $z$ as adopted in the experiment) and after (i.e., $\hat{z}$) the fusion layer simultaneously in Fig. 6. It shows that each cluster of the former is more compact compared to that of the latter (prefixed with 'f'), indicating that the fusion layer reduces the proportion of clustering-related information in the feature representation.

2. For each cluster, we calculated the singular values and singular vectors of the feature representation before and after the fusion layer. We found that the singular vector corresponding to the largest singular value is highly correlated to its corresponding cluster centroid. However, the proportion of the largest singular value among all singular values decreases a lot after the fusion step, as shown below. This indicates that the representation after fusion is less compacted and contains more sample specific details.

Table 7: The collected proportion (%) of the largest singular value among all singular values regarding each cluster using the feature representation before (i.e., $z$) and after (i.e., $\hat{z}$) the fusion layer respectively. Larger results marked in bold.

| Digit | 0 | 1 | 2 | 3 | 4 | 5 | 6 | 7 | 8 | 9 | Mean |
|---|---|---|---|---|---|---|---|---|---|---|---|
| Before Fusion | **54.81** | **52.63** | **53.65** | **64.57** | **46.45** | **57.76** | **51.16** | **53.34** | **55.94** | **29.12** | **51.94** |
| End Fusion | 40.13 | 42.29 | 27.72 | 32.33 | 31.47 | 34.60 | 33.14 | 31.07 | 27.38 | 27.21 | 32.74 |
| Difference | 14.68 | 10.34 | 25.93 | 32.24 | 14.97 | 23.16 | 18.02 | 22.27 | 28.56 | 1.91 | 19.20 |

## A.7. Experiment details

**Experiment Settings:** We implement PC-X with PyTorch [59].

- In terms of MNIST, Reuters10K, HAR, CIFAR-10 and ImageNet-10 [5], PC-X is built upon the AE architecture described in [16]. The encoder is a fully connected multi-layer perceptron (MLP) with dimensions $D$-500-500-2000-$d$. $D$ is the dimension of input and $d$ is the dimension of centroids. All layers use ReLU activation [60] except the last.

- In terms of YTF and Fashion, we adopt a convolution neural network introduced in [19]. Specifically, the encoder consists of four convolutional layers $conv(16, 3, 1, 1)$-$conv(32, 3, 2, 1)$-$conv(32, 3, 1, 1)$-$conv(16, 3, 2, 1)$ followed by a two-layer MLP with dimensions $D$-256-$d$, where $conv(16, 3, 1, 1)$ denotes a convolutional layer with channel number 16, kernel size 3, stride length 1, and padding size 1. $D$ is the dimension of the flattened CNN output and $d$ is the dimension of centroids. We apply batch normalization after each convolutional layer, followed by the ReLU activation except the last.

The decoder is mirrored of the encoder. For all datasets, the dimension of centroids is fixed to 10. The optimizer is Adam [61]. All neural network weight, as well as the clustering centroids, are initialized using a uniform distribution following [62]. The learning rate is $5 \times 10^{-4}$ and the training epoch is $1,000$. The batch size is set to 256. $\tau$ (Eq.(2)) and $\kappa$ (Eq. (6)) is set to 5 and 0.995.

**Metric:** we list the formulation of three clustering metrics, i.e., ACC, NMI and ARI, in the following:

- **Accuracy (ACC):** For sample $i$, let $R_i$ denote its ground truth label and $C_i$ be its label obtained by clustering.

$$ACC = \frac{\sum_{n=1}^{N} \delta(R_n, C_n)}{N} \times 100\%,$$

where $\delta(x, y)$ equals one if $x = y$, and zero otherwise. $N$ denotes the number of samples.

- **Normalized mutual information (NMI):** Let $R$ denote the ground truth label and $C$ be the label obtained by clustering. The NMI is defined as follows:

$$NMI = \frac{2MI(R, C)}{H(R) + H(C)},$$

where $H(X)$ is the entropy of $X$, and $MI(X, Y)$ is the mutual information of $X$ and $Y$.

- **Adjusted rand index (NMI):** Let $D \in \mathbb{R}^{K \times K}$ denote the contingency table between the ground truth label $R$ and the label $C$ obtained by clustering. The ARI is defined as follows:

$$ARI = \frac{\sum_{ij} \binom{D_{ij}}{2} - \left[ \sum_i \binom{A_i}{2} \sum_j \binom{B_j}{2} \right] / \binom{N}{2}}{\frac{1}{2} \left[ \sum_i \binom{A_i}{2} + \sum_j \binom{B_j}{2} \right] - \left[ \sum_i \binom{A_i}{2} \sum_j \binom{B_j}{2} \right] / \binom{N}{2}},$$

where $N = \sum_{ij} D_{ij}$ is the number of samples, $A \in \mathbb{R}^K = D \times \mathbb{I}_K$ is the number of samples per cluster in $R$ and $B \in \mathbb{R}^K = D^T \times \mathbb{I}_K$ is the number of samples per cluster in $C$.

For all three metrics, values range between 0 and 1, where higher value indicates better performance.

---

[5]we apply PC-X on the extracted features of CIFAR10 and ImageNet-10.

