# OpenReview forum: "PC-X: Profound Clustering via Slow Exemplars"
_CPAL.cc/2024/Conference — CPAL 2024 (Proceedings Track) Oral_

### Official Review · Reviewer_p97g · 2023-10-06
**Clear paper with some questions on experimental results**

**Rating:** 9
**Confidence:** 3

**Review:**

The authors present an intuitive clustering method that appears to be efficient and generalizable. The authors describe existing shortcomings and the motivation clearly. Although there are several grammatical errors, they are insignificant enough to not affect the clarity.

Pros
+ Thorough comparisons with various baselines and explored different data modalities
+ Nontrivial theoretical analysis

Cons
- Although the number of datasets tested is sufficient, the scale of the data is quite limited
- Included theoretical algorithm complexities but lacks empirical verification of speed-ups/memory requirements compared to other methods

Questions
1) Based on Figure 2, it is interesting that your method preserves diversity within each cluster in contrast to IDEC's embedding space where points in a cluster collapse on each other. Why is this happening, and what are the implications of this property?

2) Is there an interpretation of what the fully-connected layer does in the fusion step? Does it learn a specific way to mix information?

---

### Official Review · Reviewer_RQJF · 2023-10-08
**Method for clustering data**

**Rating:** 7
**Confidence:** 3

**Review:**

**Summary**

The authors propose a clustering method which doesn't use any domain knowledge, i.e., does not use augmentation and is interpretable, i.e., centroids can be visualized and are semantically meaningful. They propose a skip connection that encourages the interpretability of centroids and use an auxiliary variable to optimize. Results support their claim, and consistently perform better than the baselines mentioned.

**Strengths**
1. Present a simple way to encourage interpretability using the skip connection.

2. Convergence analysis, ablation study and effectiveness of different loss components are presented in the paper.

**Weakness**
1. The claim that the method is universal to any modality is incorrect. You need to change the architecture of auto-encoders for different modalities.

2. Optimization section (3.1.1.) could be improved. I'm unsure if the optimization decomposition steps are done for different batches

**Minor**
Avoid using the term 'skip connection' as it holds a particular meaning in the literature.

---

### Official Review · Reviewer_iJVS · 2023-10-15
**Review: the paper introduces a well-structured new deep clustering framework, complemented by a rich theoretical analysis and extensive experiments, but lacks a clear discussion of its limitations, and would benefit from better definitions of terminology and a more nuanced exploration of interpretability and efficiency.**

**Rating:** 7
**Confidence:** 3

**Review:**

**Quality:**
The submission is of good quality, showcasing a well-thought-out framework (PC-X) aimed at addressing challenges inherent in deep clustering. The theoretical underpinnings are strong, and the paper does a commendable job of situating PC-X within the existing landscape of clustering paradigms.

**Clarity:** The paper is well-structured and well-written, making it easy to follow. The comparative analysis using Table 1 provides a clear picture of how PC-X stands relative to existing methods, although some definitions could have been articulated better for enhanced clarity.

**Originality:**
(a) PC-X, with its skip-connection module and unique optimization algorithm, presents a novel approach to deep clustering;
(b) The idea of making clustering centroids into legible exemplars through decoding is innovative and adds a fresh perspective to the domain.

**Significance:** The paper has the potential to further the discourse in deep clustering, especially around interpretability and efficiency.

**Pros:**
- Well-structured and well-articulated paper making it easy to follow.
- Novel framework (PC-X) with innovative components like the skip-connection module and the unique optimization algorithm.
- Rich theoretical analysis backing the proposed framework. Extensive experimentation against multiple baseline methods on several datasets.

**Cons:**
- Datasets used for experimentation are not very large or diverse, which may not fully demonstrate the universality of PC-X.
- Lack of a clear discussion on the limitations of PC-X, which could have provided a more balanced view of the framework.
- Some definitions, especially that of "Universality", are not clear enough, which might lead to confusion. The discussion on interpretability and efficiency could be more nuanced, and the claims could be better substantiated.

---

### Meta-Review · Area_Chair_aDsg · 2023-11-15

**Recommendation:** Accept (Poster)
**Confidence:** 5

**Metareview:**

This paper proposes a new deep clustering algorithm trying to maintain 4 nice properties: accuracy, universality, interpretability and efficiency.  All reviewers agree the paper has novel contributions and is well-written. The convergence analysis is a nice result of the paper. The authors have tried to address the concerns of reviewers by adding more results on large-scale datasets.  Based on these, I will recommend acceptance.

---

### Decision · Program_Chairs · 2023-11-19

**Decision:**

Accept (Oral)

**Comment:**

All reviewers and AC agreed that the paper is of high quality, presenting a well-structured framework for deep clustering. It introduces novel components, like the skip-connection module and a unique optimization algorithm, and offers strong theoretical foundations. The potential to advance the field of deep clustering, particularly in terms of interpretability and efficiency, makes it a strong acceptance case.

The action PC chair for this paper is Atlas Wang, who made the decision after carefully reading the paper as well as the comments by all reviewers and AC. The decision is agreed by all PC chairs.